# Phenotypic Antimicrobial Resistance Profiles of Human *Campylobacter* Species Isolated in Northwest Italy, 2020–2023

**DOI:** 10.3390/microorganisms12030426

**Published:** 2024-02-20

**Authors:** Clara Tramuta, Aitor Garcia-Vozmediano, Cristiana Maurella, Daniela Manila Bianchi, Lucia Decastelli, Monica Pitti

**Affiliations:** 1Centro di Riferimento per la Tipizzazione delle Salmonelle, CeRTiS, Istituto Zooprofilattico Sperimentale del Piemonte Liguria e Valle d’Aosta, Via Bologna, 148, 10154 Turin, Italy; manila.bianchi@izsto.it (D.M.B.); lucia.decastelli@izsto.it (L.D.); monica.pitti@izsto.it (M.P.); 2S.S. Rischi Alimentari ed Epidemiologia degli Alimenti (REA), Istituto Zooprofilattico Sperimentale del Piemonte Liguria e Valle d’Aosta, Via Bologna, 148, 10154 Turin, Italy; aitor.garciavozmediano@izsto.it (A.G.-V.); cristiana.maurella@izsto.it (C.M.)

**Keywords:** antimicrobial susceptibility testing, *Campylobacter jejuni*, *Campylobacter coli*, foodborne pathogens, human strains, occurrence

## Abstract

The spread of antimicrobial resistant *Campylobacter* strains, linked to antimicrobials use and abuse in humans and food animals, has become a global public health problem. In this study, we determine the prevalence of antimicrobial resistance (AMR) in human *Campylobacter* isolates (n = 820) collected in Piedmont, Italy, from March 2020 to July 2023. The strains underwent susceptibility testing to determine the minimal inhibitory concentration for erythromycin, ciprofloxacin, gentamicin, streptomycin, and tetracycline: 80.1% of the strains showed resistance to at least one antibiotic. The highest prevalence of AMR was noted for ciprofloxacin and tetracycline (72.1% and 52.9%, respectively) and the lowest for erythromycin and aminoglycosides (streptomycin/gentamicin) (3.2% and 5.4%, respectively). The prevalence of co-resistance against fluoroquinolones and tetracyclines was 41.1%. The prevalence of multidrug resistant strains was 5.7%. Our data support evidence that AMR in human *Campylobacter* strains is common, particularly against ciprofloxacin and tetracycline, two medically important antimicrobials for humans.

## 1. Introduction

*Campylobacter* species are Gram-negative, spiral, or curve-shaped bacteria. These microaerophilic, non-fermentative, non-spore-forming, mobile [1] microorganisms are one of the most common causes of diarrhea worldwide. Many *Campylobacter* species are zoonotic pathogens, associated with a range of gastrointestinal diseases in humans termed campylobacteriosis [2]. Campylobacteriosis is the most commonly reported foodborne zoonosis in Europe (127,840 cases, EU notification rate of 41.1 per 100,000 of population), with a 2.1% increase in the EU notification rate compared with 2020 [3]. Active surveillance through the U.S. Foodborne Diseases Active Surveillance Network (FoodNet) reports about 20 cases of campylobacteriosis per 100,000 people diagnosed each year [4]. Since many more cases probably go undiagnosed or unreported, the CDC estimates that 1.5 million people in the United States become ill from *Campylobacter* infection every year [4].

*Campylobacter* infection is associated with gastrointestinal symptoms and inflammation of the gastrointestinal tract involving the small intestine. Symptoms are generally diarrhea. Extra-gastrointestinal complications include reactive arthritis, bacteremia, septicemia, endocarditis, and meningitis [5]. Most *Campylobacter* infections are mild and self-limiting and require only supportive therapy, while appropriate antibiotic treatment is indicated for severe or prolonged infections and in immunocompromised patients [6]. Among *Campylobacter* species, *C. jejuni* and *C. coli* are the most prevalent causative agents of gastroenteritis. Most infections in humans are correlated with food handling and the consumption of contaminated food (e.g., meat, unpasteurized milk, fruits, vegetables, or water) [7]. Poultry, domestic and wild animals are the primary reservoirs of *Campylobacter* species. The consumption of raw or undercooked poultry meat is the main risk factor for human campylobacteriosis [8,9,10].

The prevention and control of *Campylobacter* colonization in food-producing animals, such as poultry flocks, for example, involve public health strategies to reduce the incidence of campylobacteriosis in humans. Within the national surveillance networks for enteric pathogens in human medicine (Enter-NET), the regional reference center for *Salmonella* typing (Centro di Riferimento per la Tipizzazione delle Salmonelle, CeRTiS) is involved in the identification and characterization of pathogens, including *Campylobacter*. CeRTiS performs analysis of samples in official microbiological controls, investigates antimicrobial resistance (AMR) profiles, and provides support to agencies in risk analysis and epidemiological outbreak investigations. AMR surveillance of strains of human origin is carried out on a panel of molecules established by the European Centre for Disease Prevention and Control (ECDC) [11].

The antimicrobial resistance of *Campylobacter* species has increased worldwide [12] largely due to antimicrobial overuse in humans and in food-producing animals. Certain antimicrobials have been used extensively to treat *Campylobacter* infection in humans. For example, fluoroquinolones and macrolides are the drugs of choice to treat campylobacteriosis and tetracyclines and gentamicin to treat systemic infection in some cases [13]. High rates of AMR to ciprofloxacin, tetracycline, and erythromycin have been observed in *C. coli* and *C. jejuni* from human samples and food-producing animals in Europe [14].

The development of AMR has serious implications for treating *Campylobacter* infection in humans, but antimicrobial susceptibility testing can help guide appropriate therapy and monitor trends in AMR. With the present study, we wanted to determine the occurrence of phenotypic AMR of human *Campylobacter* species isolated in Piedmont in the period 2020–2023.

## 2. Materials and Methods

### 2.1. Sample Collection and Campylobacter Isolation

Isolates were collected via laboratory passive surveillance from March 2020 to July 2023. The *Campylobacter* strains originated from biological samples collected from symptomatic humans. The strains were first isolated using selective enrichment in Preston broth for 24 h at 42 °C in microaerobic atmosphere (approximately 5% O_2_, 10% CO_2_, 85% N_2_) at clinical laboratories throughout Piedmont. The isolates were then sent to the CeRTiS where they were subcultured on Columbia blood agar (Becton Dickinson, Franklin Lakes, NJ, USA) at 37 °C for 24 h and identified by MALDI TOF/TOF mass spectrometry (Bruker Daltonics GmbH, Bremen, Germany).

### 2.2. Antimicrobial Susceptibility Testing

Phenotypic testing based on the determination of minimum inhibitory concentrations (MICs) to erythromycin (1–128 µg/mL), ciprofloxacin (0.12–16 µg/mL), gentamicin (0.12–16 µg/mL), streptomycin (0.25–16 µg/mL), and tetracycline (0.5–64 µg/mL) was performed using a commercial microdilution tool (Sensititre *Campylobacter* plate–EUCAMP2, Thermo Fisher Scientific, Waltham, MA, USA) following the manufacturer’s instructions. We applied the MIC interpretive resistance standards defined by the European Committee on Antimicrobial Susceptibility Testing (EUCAST) and EFSA [15,16] to define isolates of *C. jejuni* resistant to erythromycin > 4 µg/mL, ciprofloxacin > 0.5 µg/mL, tetracycline > 2 µg/mL, streptomycin > 4 µg/mL or gentamicin > 2 µg/mL. *C. coli* was defined resistant when the MIC equated to erythromycin was > 8 µg/mL, ciprofloxacin > 0.5 µg/mL, tetracycline > 2 µg/mL, streptomycin > 4 µg/mL or gentamicin > 2 µg/mL. The MIC ranges, MIC_50_ and MIC_90_, were calculated separately for each species; MIC_50_ and MIC_90_ represented the antibiotic concentrations (µg/mL) at which 50% and 90% of the *Campylobacter* isolates could be inhibited, respectively. Multidrug resistance (MDR) was defined as resistance to at least three antimicrobial classes.

### 2.3. Statistical Analysis

Prevalence of AMR was calculated as the percentage of microbial strains exhibiting resistance to at least one antibiotic, together with corresponding exact 95% confidence intervals (95% CI), for each *Campylobacter* species and by antibiotic. Differences in AMR prevalence were evaluated using a Chi-squared test, while the cumulate number of AMR per isolate was modelled with Poisson regression, considering the *Campylobacter* species as a covariate. The prevalence ratio (PR) was used to express the results of the model. Data analysis was performed using STATA 17 [17] and the statistical significance level was set at 5%.

## 3. Results

Between 2020 and 2023, the regional laboratory surveillance system identified 820 cases of *Campylobacter* infection in humans. The leading causative agents were *C. jejuni* and *C. coli*, with *C. jejuni* accounting for 87.7% of cases. The strains were collected from patients with gastrointestinal symptoms (n = 802), septicemia (n = 17) or urinary tract infection (n = 1).

AMR was frequent: 80.1% of the strains were resistant to at least one antibiotic. AMR was highest against ciprofloxacin and tetracycline (prevalence of 72.1% and 52.9%, respectively) and lowest against erythromycin (3.2%) and aminoglycosides (streptomycin/gentamicin) (5.4%). The distribution of MICs, MIC_50_ and MIC_90_ of the *Campylobacter* isolates against five common antibiotics are shown in Table 1.

The frequency of AMR observed in *C. jejuni* strains (79.6%; 95% CI 76.4–82.4) was comparable to that observed in *C. coli* strains (84.2; 95% CI 75.6–90.7; Chi-squared test, *p* > 0.05). However, differences in the occurrence of AMR against single antibiotics were observed, especially against erythromycin and tetracycline for which *C. coli* displayed higher AMR levels than those observed in *C. jejuni* isolates (Figure 1). In addition, differences were noted in AMR against gentamycin and streptomycin: the prevalence in *C. coli* (7.9%; 95% CI 3.5–15.0) was higher than that observed against gentamycin in *C. jejuni* (0.1%; 95% CI 0.004–0.8; *p* < 0.001), while no differences were noted between the two *Campylobacter* species against streptomycin (7.9% in *C. coli* vs. 4.3% in *C. jejuni*, *p* > 0.05).

*C. coli* strains were more likely than *C. jejuni* strains to exhibit co-occurring AMR against two or more antimicrobial classes (PR 1.32; 95% CI 1.15–1.51). The most common co-resistance pattern was resistance against two antimicrobial classes, recorded for 41.1% (n = 337) of *Campylobacter* strains. Within this pattern, co-resistance against fluoroquinolones and tetracyclines was the most prevalent combination (Table 2). The profiles involving macrolides with tetracyclines and aminoglycosides and fluoroquinolones with aminoglycosides were found only in *C. coli* (n = 3). Furthermore, we noted a prevalence of 5.7% for MDR strains circulating in the study area and most commonly involving fluoroquinolones. Concurrent resistance against all four antimicrobial classes was identified only in *C. coli* strains, which accounted for 10.6% of the total number of resistant strains identified.

## 4. Discussion

Since 2005, *Campylobacter* has become the most commonly reported cause of bacterial food-borne illness in the European Union [10]. Because it is a zoonotic pathogen, it is exposed to antibiotics for both human and veterinary medicine. For example, fluoroquinolones, which are critically important antimicrobials (CIAs), and tetracycline have been used over the past 50 years to promote growth and to treat infection in poultry [13]. Since campylobacteriosis is generally a self-limiting illness, it is not usually treated. In contrast, macrolides and fluoroquinolones are the antibiotics of choice in the treatment of severe or persistent illness [18]. Antibiotic resistance of *Campylobacter* to these classes of antibiotics, especially fluoroquinolones, continues to increase. For these reasons, *Campylobacter* has been identified as a public health threat by both the World Health Organization (WHO) and the U.S. CDC [19]. The increase in *Campylobacter* strains resistant to common antibiotics highlights the need for improved surveillance and data sharing.

Resistance to the empirical drugs erythromycin, ciprofloxacin, and tetracyclines has been reported for human clinical strains in many countries worldwide: a previous study in Quebec (Canada) showed a 50% tetracycline resistance among *C. coli* isolates and 39% among *C. jejuni* isolates [20] and another study in South Africa recorded resistance to ciprofloxacin and erythromycin in 33.3% and 38.9% of *C. coli* and 20% and 31.5% of *C. jejuni*, respectively [21].

Our data show high resistance to ciprofloxacin (72.1%) for the period between 2020 and 2023 in Piedmont. These findings are shared by previous reports for Italy: high levels of resistance in *C. jejuni* and *C. coli* to ciprofloxacin (76% and 70%, respectively) from human samples [13,22]. These rates are consistent also with those reported for the EU between 2019 and 2021, where ciprofloxacin resistance in *Campylobacter* isolated from human samples was high to extremely high (range, 22.2% to 100% for *C. jejuni* and for *C. coli*). Very high levels of resistance, higher for *C. coli* than for *C. jejuni*, were observed for ciprofloxacin also in isolates from food-producing animals (range, 41.7% to 80.4%) [14,23]. Moreover, the high resistance to tetracycline (52.9%) we observed is consistent with data for the EU: 45.3% in *C. jejuni* and 70.3% in *C. coli* from human isolates. The resistance to tetracycline ranged from high to extremely high (43.3–90.5%) also in food-producing animals [14]. Antimicrobial resistance to aminoglycosides (gentamicin and streptomycin) was less frequent (5.4%), roughly similar to previous studies where the resistance to gentamicin and streptomycin was low (0.7% and 2.4% on average, respectively) in *C. jejuni* and *C. coli* isolates from human samples [13,14]. Conversely, we observed low rates of resistance to erythromycin (3.2%), another CIA, supporting previous data that indicated low resistance in *C. jejuni* from human and animal samples [12,13,14]. However, higher rates have been reported for *C. coli* isolates from human samples (8.5%), similar to our findings, and animals (range, 4.4% to 35.7%) [14].

The high rate of co-occurrence of resistance to ciprofloxacin and tetracycline in *C. jejuni* (89.9%) raises concerns for public health. Lower percentages were reported by a *Campylobacter* spp. surveillance study performed in the 2013–2016 period by the Enter-Net Italia network where the same co-occurrence of resistance was observed in 48% of *C. jejuni* and 41% of *C. coli* (total n = 647) [22]. The higher resistance we observed could be due to the many intensive poultry farms operating in our area. Indeed, *Campylobacter* spp. isolates from slaughtered poultry in northern Italy demonstrated high resistance to quinolones, tetracycline, and macrolides [22]. This high prevalence of co-resistance underscores the importance of monitoring for resistant *Campylobacter* spp. strains in food-producing animals and the transmission to humans through the food chain.

Regarding multidrug resistance (MDR), co-resistance to ciprofloxacin, tetracycline, and aminoglycosides was markedly higher in *C. jejuni* (96.2%) than in *C. coli*, In contrast, the highest rate of co-resistance to ciprofloxacin, tetracycline, and erythromycin (92.3%) was noted for *C. coli* although the number of isolates was not high; lower rates of MDR to these three antimicrobials were previously reported for *C. coli* of human origin (29%) [22].

## 5. Conclusions

The present study provides evidence that AMR is common among human *Campylobacter* strains isolated in the study area, particularly against ciprofloxacin and tetracycline. The further emergence of *Campylobacter* resistance may be prevented through the implementation of good antimicrobial stewardship at the farm level, since *Campylobacter* can easily reach the consumer via the food production chain and pose a serious public health risk. Moreover, clinicians should consider optimal treatment before beginning empiric treatment and use antibiotics judiciously for the protection of their patients and the community.

## Figures and Tables

**Figure 1 microorganisms-12-00426-f001:**
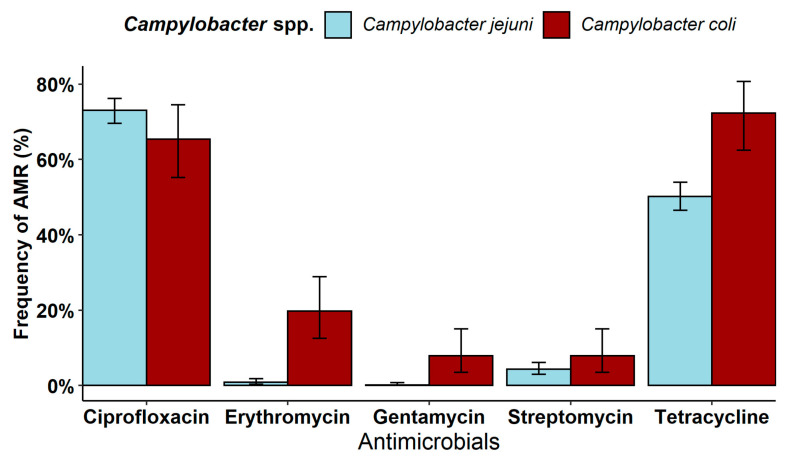
Antimicrobial resistance (95% confidence interval) of *Campylobacter* spp. against antibiotics tested.

**Table 1 microorganisms-12-00426-t001:** Distribution of MICs, MIC_50_ and MIC_90_ of human *Campylobacter* species against five common antibiotics.

	*C. jejuni*			*C. coli*		
Antibiotic	MIC Range (µg/mL)	MIC_50_ (µg/mL)	MIC_90_ (µg/mL)	MIC Range (µg/mL)	MIC_50_ (µg/mL)	MIC_90_ (µg/mL)
Erythromicin	1–128	≤1	2	1–128	≤1	128
Ciprofloxacin	0.12–16	4	8	0.12–16	4	8
Tetracycline	0.5–64	4	32	0.5–64	32	64
Streptomycin	0.25–16	0.5	1	0.25–16	1	16
Gentamicin	0.12–16	≤0.125	0.25	0.12–16	0.25	1

**Table 2 microorganisms-12-00426-t002:** Co-resistance profiles of human *Campylobacter* species.

Co-Resistance	*Campylobacter* Species	N° of Strains	Resistance Profile	N° of Strains	% of Strains
Resistant to two antimicrobial classes	*C. jejuni*	327	Fluoroquinolones Tetracyclines	291	89.9
	*C. coli*			36	11.0
	*C. jejuni*	4	Aminoglycosides Tetracyclines	2	50.0
	*C. coli*			2	50.0
	*C. jejuni*	3	Fluoroquinolones Macrolides	1	33.3
	*C. coli*			2	66.7
	*C. jejuni*	1	Macrolides Tetracyclines	0	0
	*C. coli*			1	100
	*C. jejuni*	1	Aminoglycosides Macrolides	0	0
	*C. coli*			1	100
	*C. jejuni*	1	Fluoroquinolones Aminoglycosides	0	0
	*C. coli*			1	100
Resistant to three antimicrobial classes	*C. jejuni*	26	Fluoroquinolones Aminoglycosides Tetracyclines	25	96.2
	*C. coli*			1	3.8
	*C. jejuni*	13	Fluoroquinolones Macrolides Tetracyclines	1	7.7
	*C. coli*			12	92.3
	*C. jejuni*	1	Fluoroquinolones Macrolides Aminoglycosides	1	100
	*C. coli*			0	0
Resistant to four antimicrobial classes	*C. jejuni*	7	Fluoroquinolones Macrolides Aminoglycosides Tetracyclines	0	0
	*C. coli*			7	100

## Data Availability

Data are contained within the article.

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
