# Peer review of "Phenotypic Antimicrobial Resistance Profiles of Human Campylobacter Species Isolated in Northwest Italy, 2020–2023"

_microorganisms, 2024, doi:10.3390/microorganisms12030426_

Round 1

Reviewer 1 Report

Comments and Suggestions for Authors

This study reports on the prevalence of resistance among human isolates of Campylobacter spp. in north-western Italy over the last four years. It provides interesting data on the phenotypic resistance of the tested isolates to five antibiotics (ciprofloxacin, tetracycline, erythromycin, streptomycin and gentamicin).

The manuscript is appropriately structured, and the format follows the requirements in the Instructions for Authors in Microorganisms. Appropriate methods were used to achieve the objectives of the study and the results are treated statistically. Although this is a "Communications" manuscript, the manuscript is very brief and I recommend the following edits and additions before publication.

Minor comments:

1. Abstract - line 23: add "animal" or "poultry" farms.

2. Keywords: replace "antimicrobial resistance" with "antimicrobial susceptibility testing".

Major comments:

1. Material and Methods section:

2.1. Sample collection - add some additional information about the isolates included in the study - for example, whether or not some isolates or even a larger group of isolates could have come from the same source of infection, how the same source of infection could affect the results of resistance monitoring - the likelihood of including the same strains in monitoring. Has this been taken into account in any way? In what way?

2.2 Antimicrobial susceptibility testing - EUCAST does not provide breakpoints for Campylobacter spp. for streptomycin and gentamicin. From what source were these breakpoints taken or how were they defined? Please indicate.

2. Results section:

The results are commented mostly verbally, the number and percentage of resistant isolates to the tested antibiotics are given. I recommend adding to the results (e.g. in the form of a simple table) the distribution of minimum inhibitory concentrations (MICs) found for each antimicrobial tested. This information, which is certainly available to the authors, may increase the value of the published results to the reader. In addition, MIC50 and MIC90 values could be provided that may suggest the trends in the development of resistance of the isolates tested to the individual antimicrobials (Schwarz, S.; Silley, P.; Simjee, S.; Woodford, N.; van Duijkeren, E.; Johnson, A.P.; Gaastra, W. Assessing the antimicrobial susceptibility of bacteria obtained from animals. Vet. Microbiol 2010, 141, 601–604.)

Author Response

Dear referee,

We resubmit our manuscript entitled “Phenotypic Antimicrobial Resistance Profiles of Human Campylobacter species isolated in Northwest Italy in the last four years (2020-2023)”.

Thank you for your revision and the suggestions, that improved the quality of the manuscript. Table below gives a point-by point reply to comments.

Best regards, Clara Tramuta, corresponding author

Referee 1

This study reports on the prevalence of resistance among human isolates of Campylobacter spp. in north-western Italy over the last four years. It provides interesting data on the phenotypic resistance of the tested isolates to five antibiotics (ciprofloxacin, tetracycline, erythromycin, streptomycin and gentamicin).

The manuscript is appropriately structured, and the format follows the requirements in the Instructions for Authors in Microorganisms. Appropriate methods were used to achieve the objectives of the study and the results are treated statistically. Although this is a "Communications" manuscript, the manuscript is very brief and I recommend the following edits and additions before publication.

Note

Answer

Lines in the revised paper

1. Minor comments

Abstract - line 23: add "animal" or "poultry" farms.

Clarified.

Line 23.

2. Minor comments

Keywords: replace "antimicrobial resistance" with "antimicrobial susceptibility testing".

Clarified.

Line 26.

Major comments

1. Material and Methods section:

2.1. Sample collection - add some additional information about the isolates included in the study - for example, whether or not some isolates or even a larger group of isolates could have come from the same source of infection, how the same source of infection could affect the results of resistance monitoring - the likelihood of including the same strains in monitoring. Has this been taken into account in any way? In what way?

The strains were isolated by several clinical laboratories and no evidence about outbreaks were given; no correlations between infection sources and clinical cases were investigated.

Major comments

1. Material and Methods section:

2.2. Antimicrobial susceptibility testing - EUCAST does not provide breakpoints for Campylobacter spp. for streptomycin and gentamicin. From what source were these breakpoints taken or how were they defined? Please indicate.

Sections material and methods and references provides information: EFSA (European Food Safety Authority), Aerts M. et al., 2019. Scientific report on the technical specifications on harmonised monitoring of

antimicrobial resistance in zoonotic and indicator bacteria from food-producing animals and food. EFSA

Journal 2019;17(6):5709, 122 pp. https://doi.org/10.2903/j.efsa.2019.5709

Line 96.

Major comments

2. Results section:

The results are commented mostly verbally, the number and percentage of resistant isolates to the tested antibiotics are given. I recommend adding to the results (e.g. in the form of a simple table) the distribution of minimum inhibitory concentrations (MICs) found for each antimicrobial tested. This information, which is certainly available to the authors, may increase the value of the published results to the reader. In addition, MIC50 and MIC90 values could be provided that may suggest the trends in the development of resistance of the isolates tested to the individual antimicrobials (Schwarz, S.; Silley, P.; Simjee, S.; Woodford, N.; van Duijkeren, E.; Johnson, A.P.; Gaastra, W. Assessing the antimicrobial susceptibility of bacteria obtained from animals. Vet. Microbiol 2010, 141, 601–604.)

A table provide details about the distribution of MICs.

Details are given in the text.

Lines 100-103, 124-125, 127-130.

Reviewer 2 Report

Comments and Suggestions for Authors

Check the entire manuscript...minor reviosions

Comments on the Quality of English Language

This work is very interesting and correctly written...Only some minor revisions:

lane 44...deletion of some part?

Figure 1: delete ***

Table 1: improve it, too little and not understandable

Author Response

Dear referee,

We resubmit our manuscript entitled “Phenotypic Antimicrobial Resistance Profiles of Human Campylobacter species isolated in Northwest Italy in the last four years (2020-2023)”.

Thank you for your revision and the suggestions, that improved the quality of the manuscript. Table below gives a point-by point reply to comments.

Best regards, Clara Tramuta, corresponding author

Referee 2

Comments and Suggestions for Authors: Check the entire manuscript...minor revisions.

Comments on the Quality of English Language: This work is very interesting and correctly written...Only some minor revisions:

Note

Answer

Lines in the revised paper

1

lane 44...deletion of some part?

Deleted.

Lines 44-45.

2

Figure 1: delete ***

Deleted.

Lines 140-141.

3

Table 1: improve it, too little and not understandable

More details are now given in Table (now Table 2).

Lines 158-159.

Round 2

Reviewer 1 Report

Comments and Suggestions for Authors

I thank the authors for accepting my comments and for revising the manuscript.

Author Response

Thank you for the useful suggestions.

best regards,

the authors